# Lower Urinary Tract Infections: An Approach for Greek Community Health Practitioners

**Diamantis Klimentidis** [1] and **Georgios Pappas** [2,*]

1   Psychiatric Clinic Agia Aikaterini, 57010 Thessaloniki, Greece; diamantis.klimentidis@gmail.com
2   Institute of Continuing Medical Education of Ioannina, 45333 Ioannina, Greece
*   Correspondence: gpele@otenet.gr; Tel.: +30-265-102-8289

**Abstract:** Urinary tract infections (UTIs) cause a significant infectious burden in the community and demand a coordinated approach from all first-line health professionals. Uncomplicated UTIs refer to infections in non-pregnant females without any underlying localized or generalized pathology, diagnosed through history by the presence of specific local symptoms and in the absence of systemic ones like fever. Uncomplicated UTIs are usually caused by *Escherichia coli* species; thus, empirical antibiotic treatment can be immediately initiated. A percentage of patients can experience a resolution of symptoms without therapy; however, this "wait and observe" approach is supported only by the relevant British guidelines. There are limited quality studies in the literature on adjuvant treatment options; these can include BNO 145, a phytotherapeutic medicine, and XHP, a medical technology product. Despite being licensed by the European Medicines Agency on the basis of traditional use, there is inadequate support in the medical literature for the use of cranberry extracts and *Arctostaphylos uva-ursi* extracts. The use of antibiotics is associated with higher response rates and urine culture sterilization as well as lower recurrence/relapse rates; on the other hand, side effect rates are also higher. In choosing the proper empirical antibiotic therapy, one has to take into account individual patient characteristics and community resistance patterns as well as the antimicrobial resistance pressure exerted by the wide use of a specific antibiotic. There is a need for a common framework through which all frontline health practitioners should operate when faced with a case of uncomplicated UTI. In Greece, there are three different guidelines for UTI treatment, developed by the Ministry of Health, the National Organization for Medicines, and the Hellenic Society for Infectious Diseases. The authors of the present study aim at synthesizing these guidelines as well as relevant guidelines from international scientific or other national regulatory organizations while taking into account local resistance patterns. The authors propose the first-line use of either fosfomycin, nitrofurantoin, or pivmecillinam. The use of trimethoprim/sulfamethoxazole is discouraged due to increased resistance of Greek community *E. coli* isolates. Fluoroquinolone use should be avoided due to high *E. coli* community resistance (exceeding 20% for Greece), along with their unfavorable benefit/side effect balance in uncomplicated UTIs, as well as the overall community resistance pressure exerted by their use. A 5-day regimen remains superior to a 3-day one; the latter may be suitable for certain, not yet adequately characterized, patients.

**Keywords:** urinary tract infections; guidelines; nitrofurantoin; fosfomycin; pivmecillinam; resistance

## 1. Introduction

Urinary tract infections (UTIs) are among the commonest community-acquired infections, affecting more than 50% of females during their lifetime, of whom 20–40% may experience recurrent infections. Annually, 11% of females experience symptoms of a lower UTI, and it has been estimated that more than 30% of females will have received an antibiotic prescription for a UTI by the age of 24 [1,2]. Hence, awareness about their clinical presentation, the diagnostic approach, differential diagnosis, and optional treatment is of paramount importance for all levels of community health professionals: females with

UTI may seek primary advice from general practitioners or physicians, from clinical microbiologists, or from pharmacists. Regarding the latter, one should note that community pharmacists are often the primary source of information on symptoms compatible with a UTI, and that certain countries, such as Canada, Australia, and the United Kingdom, allow community pharmacists to handle cases of a lower UTI from diagnosis to follow-up with no further intervention from other health practitioners [3–5]. Regardless of the specific health professional first encountering a patient with a novel lower UTI, this particular health professional should be able to recognize the complicated cases that are in need of a further specialized diagnostic and therapeutic approach; they should also be capable of a rapid diagnosis of an uncomplicated case, avoiding referral to unnecessary diagnostic workup and ensuring that proper therapeutic antibiotic choices are offered to the patient. Erroneous antibiotic therapeutic choices may include over-zealous antibiotic choices, antibiotics not expected to be active against the common UTI-causing pathogens, antibiotics for which commonly implicated microorganisms show significant resistance in the specific community, errors in the dose or duration of treatment, no dose modification when appropriate as in chronic renal failure, and non-evaluation of potential drug–drug interactions.

Ideally, this accurate diagnosis and patient orientation towards further workup or treatment should be performed by all community health professionals that may be the first to advise a patient; moreover, it should be performed through a common guideline framework, one that is continuously updated, adjusting, for example, to changing local microorganism resistance patterns. In the present article, we aimed to evaluate how such a common guideline framework could be applied in Greek community settings based on up-to-date literature data regarding diagnosis, treatment, and prevention; on existing international guidelines on UTIs; and on existing Greek guidelines, either from health-related state authorities or relevant professional medical associations. Based on these, we built an optional algorithm that will allow frontline health professionals to adequately recognize an uncomplicated UTI, to acknowledge factors that make a UTI complicated, to differentially diagnose it from asymptomatic bacteriuria, and to administer or approve or advise on the proper antimicrobial therapy for the proper duration of treatment. In this framework, the health professional serves not only in the best interest of the patient, but also in the best interest of the community, by salvaging antimicrobials that exert antimicrobial resistance-generating pressure through limiting their wider use.

## 2. Epidemiological Basics

The vast majority of UTIs are caused by pathogens that typically reside in the gastrointestinal lumen and can enter by proximity to the urethra and bladder and, in cases of upper UTIs, ascend through the ureters to the kidneys. The anatomy of the female urethra renders women significantly more vulnerable to UTIs compared to men. The latter's anatomy serves, under normal circumstances, as an obstacle to the colonization of the urinary tract by gastrointestinal pathogens, with a potential antibacterial role also suggested for the prostatic fluid [6].

Beyond anatomy, there are other risk factors for the development of a UTI, as recognized by the European Association of Urology (EAU) [7]. For pre-menopausal females, these include sexual contact, spermicidal use, recent changes in sexual partners, a maternal history of UTIs, history of UTIs during childhood, as well as certain blood types (the latter correlation, though, is of questionable significance). For post-menopausal females, further risk factors include a relevant pre-menopausal history, vaginal atrophy, urinary incontinence, and ultrasonographically confirmed residual post-void urine retention. For post-menopausal females residing in community care settings, indwelling bladder catheterization and diminished cognitive function also serve as risk factors for UTI development [8].

## 3. Definitions

Although there is significant heterogeneity in defining UTI forms in the literature and clinical studies [9], the EAU categorization [7] can be considered the gold standard.

### 3.1. An Uncomplicated Lower UTI

An uncomplicated lower UTI (uncomplicated cystitis) is defined by the development of more than one localized symptom in a previously healthy female. Different scientific bodies' guidelines use different symptom combinations for the definition of a UTI. Thus, a proposed combination includes the presence of two or more of the following: increased urinary frequency, dysuria, urinary urgency, and sense of lower abdominal/suprapubic discomfort; this combination does not take into account the presence of cloudy or foul-smelling urine. A different proposed diagnostic framework, though, demands the presence of two or more of the following: dysuria, frequent nocturnal urination, and cloudy urine. Finally, another considers isolated dysuria as a diagnostically adequate symptom, adding that in its absence, in order to diagnose a UTI, one needs the presence of two or more of the following: hematuria, lower abdominal/suprapubic discomfort, and urgency to urinate [3–5,7,10]. Irrespective of the initial symptom combination used for UTI diagnosis, in order to characterize a UTI as lower, one needs the absence of fever and constitutional symptoms and of generalized abdominal or lumbar pain (although these symptoms may be difficult to evaluate in older patients with cognitive impairment).

### 3.2. A Complicated Lower UTI

A complicated lower UTI refers to any lower UTI observed in males, pregnant females, females with underlying anatomical lesions of the genitourinary tract, and females with an underlying immune-compromising condition (which may be generalized immunodeficiency caused by cancer or post-transplantation medications or a consequence of simpler, common morbidities; of these, diabetes mellitus is of particular importance, given the increasing use of sodium-glucose cotransporter 2 (*SGLT*-2) inhibitors in its treatment, which further predispose to the development of UTIs on top of diabetes-induced immune deficiency).

### 3.3. An Upper UTI

An upper UTI, often called pyelonephritis (although the term also applies to non-infectious conditions), is defined by the additional presence of fever, abdominal or lumbar pain, and constitutional symptoms indicative of bacteremia (chills, rigor, vomiting).

### 3.4. Asymptomatic Bacteriuria

Asymptomatic bacteriuria is defined as the presence of microbes in a urine culture in the absence of any symptom, in a concentration of at least $10^5$/mL (or $10^3$/mL for males), demonstrated at least twice, in cultures taken at least 24 h apart. As already mentioned, though, the term "absence of any symptom" can be debatable in elderly individuals.

### 3.5. Relapse of an Uncomplicated UTI

Relapse of an uncomplicated UTI is defined as the re-appearance of the initial symptoms during the first 8 weeks after initial infection, one that can be safely attributed to the persistence of the initially implicated pathogen due to eradication failure (this definition is based on the guidelines of the Hellenic Society for Infectious Diseases [11]). Any novel appearance of symptoms compatible with UTI post the 8-week period should be considered a re-infection.

### 3.6. Chronic Recurrent UTI

Chronic recurrent UTI is defined as the appearance of at least three episodes of lower UTI in the chronic period of a year or at least two episodes in a period of 6 months, as defined by the EAU.

## 4. Etiology

The vast majority of UTIs, in percentages exceeding 75%, are caused by *Escherichia coli* strains [8,12,13]. Less frequently identified pathogens (observed more commonly in complicated UTIs) include *Staphylococcus* species (when isolated, one needs to differentiate

from contaminants during urine specimen collection), *Proteus mirabilis*, and *Klebsiella* and *Enterococcus* species. *Pseudomonas* and *Candida* species can also cause UTIs, particularly in situations related to biofilm formation, indwelling bladder catheterization, or hospital-acquired infections.

## 5. Diagnostic Approach

### 5.1. Step 1 (Is It a UTI?)

Step 1 (Is it a UTI?) in the diagnostic approach to a patient presenting with symptoms that could be attributed to a UTI is the exclusion of another diagnosis, a process that can largely achieved through patient history, aided, in selected cases, by paraclinical tests: for example, the presence of hematuria and abdominal pain necessitates the exclusion of renal/ureteral lithiasis through ultrasonography or computerized tomography (less confidently through simple X-rays). Similarly, the presence of suprapubic discomfort, when accompanied by vaginal secretions, orientates towards a diagnosis of vaginal candidiasis. On the other hand, polyuria, which may be a hallmark of systematic pathology, such as, for example, undiagnosed diabetes mellitus, may be erroneously considered as increased urinary frequency orientating towards the diagnosis of a non-existent UTI and delaying the diagnosis of the more serious underlying pathology [14]. Differential diagnosis should further take into account less common and more specific causes of similar or overlapping symptoms, such as sexually transmitted infections, urethritis, bladder cancer, aseptic cystitis (such as hemorrhagic cystitis occurring as a complication of specific chemotherapeutic regimens or post radiation), and pelvic inflammatory disease. Atypical cases of appendicitis or pressure phenomena caused by ovarian cysts may also exhibit an overlapping symptomatology. Even benign situations, such as changes in urine color due to natural ingredients/food (beetroot being a typical example) or medications (rifampicin use being an example), can often be erroneously considered indicative of a urinary tract pathology by the patient [15,16].

### 5.2. Step 2 (Is It Uncomplicated?)

Step 2 (Is it uncomplicated?) can readily be clarified through patient history: the patient's sex and underlying medical history, as well as pregnancy for females, can suggest a complicated UTI, while the described symptomatology (such as the presence of fever and constitutional symptoms) can orientate towards a diagnosis of an upper UTI. Both complicated lower UTIs and upper UTIs necessitate evaluation by a specialist, typically accompanied by further paraclinical workup and, in certain cases, hospitalization.

### 5.3. Step 3 (Do We Need Further Tests?)

Step 3 (Do we need further tests?) of the diagnostic approach may be warranted in cases of vague symptomatology that is not compatible with a definite clinical diagnosis of an uncomplicated UTI. A urine dipstick test is the commonest, first-in-line diagnostic approach: one needs to evaluate blood, nitrites, or leukocyte esterase positivity. Urine dipstick tests have a significant negative prognostic value [17] (i.e., can safely inform you on whether certain symptoms are NOT due to UTI), particularly when negative both for leukocyte esterase and nitrites. Nitrite positivity is considered highly specific for UTI diagnosis, while isolated leukocyte esterase positivity is of varying specificity [16]. pH values, determined through the dipstick test, can orientate us towards particular pathogens causing a UTI, a typical example being the correlation of alkaline pH values with *P. mirabilis* infections. One should note though that in the presence of a UTI-compatible symptom combination and the absence of factors indicative of a complicated or upper UTI, there is no need of performing either a urinary dipstick test or any other diagnostic procedure. Regarding urine culture in particular, it has been demonstrated that when all three of dysuria, increased urinary frequency, and cloudy urine are present, the likelihood of a confirmed positive urine culture is 82% (similarly, it is 74% when two of the aforementioned symptoms are present and 68% with only one of the symptoms) [18]. Yet, given that the

vast majority of uncomplicated UTIs are caused by *E. coli*, there is no rational reason to perform a urine culture, apart from cases of recurrent UTIs.

## 6. Therapeutic Approaches beyond Antibiotics

### 6.1. Do All Uncomplicated UTIs Need Treatment?

A percentage of symptoms that can be attributed to uncomplicated UTIs can self-resolve (or resolve through "unconventional" approaches); this percentage reaches 42% in the available literature [19,20]. The British National Institute for Healthcare and Excellence guidelines on UTIs underline that, in the case of uncomplicated UTI a 48 h "wait and observe" approach without any therapeutic intervention, apart from symptom relief, can be considered a valid approach for selected patients [10].

### 6.2. Is There a Role for Non-Pharmaceutical or Over-the-Counter Therapeutic Interventions?

Such approaches may be complementary to antibiotic prescription for symptom alleviation or used in isolation during a "wait and observe" period. It should be stressed, however, that these are not definite therapeutic approaches, and when suggested by a frontline healthcare professional, they should be accompanied by clarification of their adjuvant role and a strong reminder that persistence of symptoms beyond 48 h will necessitate further treatment.

#### 6.2.1. Increased Hydration

Increased hydration has been limitedly evaluated as a means of symptom alleviation in uncomplicated UTI. Increased urine flow can enhance the washing out of bacteria from the urinary tract; there are inherent difficulties, though, in designing a proper study evaluating the efficacy of increased hydration [21]: accurately estimating total hydration may be hard, and "adequate" hydration is, moreover, a subject affected by individual biometric patient characteristics (which, if ignored, may predispose to the risks of over-hydration).

#### 6.2.2. Urine Alkalinization

Urine alkalinization is not an approach supported by the existing literature on UTI therapeutics; furthermore, it can negatively affect the antibacterial efficacy of nitrofurantoin, a first-choice antibiotic that is ineffective at pH values above 8.5 [22].

#### 6.2.3. Non-Steroidal Anti-Inflammatory Drugs (NSAIDs)

Non-steroidal anti-inflammatory drugs (NSAIDs) have been repeatedly evaluated in clinical practice [23–25], since many UTI symptoms can be attributed to the host inflammatory response to pathogen invasion of the urinary tract. Most studies have been performed with ibuprofen, and their characteristics and quality vary: a higher proportion of complications was observed in two studies comparing ibuprofen with an antibiotic, with progression to pyelonephritis in the ibuprofen arm [24,25] or gastrointestinal complications, including hemorrhage [24].

Interventions based on methenamine, d-mannose, and probiotics have mainly been evaluated in terms of recurrence prevention and not treatment of acute infections and are discussed in the relevant section.

### 6.3. Phytotherapeutics, Nutritional Supplements, and Medical Technology Products

In Europe, medicinal products of herbal origin are regulated by the Committee on Herbal and Medicinal Products on account of European Medicines Agency (EMA), resulting from the development of two monograph categories based on data on safety, effectiveness, and historical information: "Well-established use" and "Traditional use". For the former, there are sufficient safety and efficacy data, whereas for the latter, there are sufficient safety data, an adequate theoretical mode of action, but no efficacy data (what is called plausible efficacy) [26].

Phytotherapeutics have certain particularities that need be addressed: first, their pharmacological action is based in one or more of their natural constituents, being on par with drugs. On the other hand, they are typically based on extracts that may include

numerous active ingredients, the majority of which have been minimally studied regarding their modes of action and interactions.

Phytotherapeutic products that are marketed as nutritional supplements are not regulated with the same vigilance that is applied to medicines. Thus, there is an inherent insecurity regarding their qualitative and quantitative properties. It is up to any health professional's scientific integrity to judiciously advice for or against the use of such products (for any indication, not limited to UTI), and evidence from the literature should be considered a pre-requisite. One can feel safer, on the other hand, regarding the proper use of phytotherapeutics that have been registered as medicines.

Medical technology products do not exert a direct medicinal or immune-modulating action in, or on, the human body, or through an intervention in human metabolism, but their mode of action can indirectly interfere with them [27].

### 6.3.1. BNO 1045

BNO 1045, marketed as Canephron, is a synthetic phytotherapeutic medicine of "Traditional use" that in each of its capsules (which is how it is sold) includes 36 mg of each of the following: centaury herb (*Centaurium erythrae*), lovage root (*Levisticum officinale*), and rosemary leaf (*Rosmarinus officinalis*). Its action is analgesic, anti-inflammatory, and anti-adhesive (against *E.coli*'s adhesion to bladder epithelial cells). Its efficacy as an adjuvant aiming at symptom alleviation, as a primary therapeutic approach, and also in minimizing symptomatic recurrences has been demonstrated in randomized and retrospective studies [28–30]. Its safety profile is excellent [31] and it can thus be considered both a useful adjuvant option and a first option in cases where one decides on a "wait and observe" approach; it should be noted, though, that in certain countries, including in Greece, it is approved as an adjuvant and not as a therapeutic option at present. On the other hand, it is approved as a phytotherapeutic medicine and not as a nutritional supplement, further confirming its safety profile.

### 6.3.2. Cranberry

Cranberry extracts have been historically popular in the treatment of lower urinary tract pathologies. The EMA monograph classifies them as a product of "Traditional use" that can be used both for recurrence prevention and for symptom alleviation in acute infection [32]. There is ample bibliographic support for the former indication: according to a recent Cochrane meta-analysis, cranberry extract can aid in preventing recurrences of UTIs in females with frequent episodes, in children, and in individuals who have undergone surgical interventions involving the bladder [33]. On the other hand, there are no clinical data that may support its use in the acute phase of the infection [34]. In Greece, cranberry extracts are marketed as nutritional supplements, with the resulting potential regulation gaps discussed above.

### 6.3.3. *Arctostaphylos uva-ursi* Extract

*Arctostaphylos uva-ursi* extract is another phytotherapeutic medicine of "Traditional use", its indications including the alleviation of symptoms in mild lower UTIs in females. According to the latest update of the EMA monograph, its suggested mechanism of action is based on the in vitro antimicrobial properties of its extract; the complete absence of any clinical efficacy data is duly noted, though [35]. Two randomized clinical trials have been performed since: the first failed to notice any efficacy against placebo [36], while the second compared the extract to fosfomycin, resulting in a significantly higher symptom burden and more complications in the extract group [37]. In Greece, *Arctostaphylos uva-ursi* extract is marketed as nutritional supplement (see above for the potential drawbacks).

### 6.3.4. XHP (Xyloglucan, Hibiscus, and Propolis)

XHP (xyloglucan, hibiscus, and propolis) is a medical technology product containing, in its form marketed in Greece, 100 mg of each of the three ingredients. It has been shown

to have positive effects in a randomized controlled trial for the treatment of lower UTIs but as an adjuvant to antibiotic treatment [38,39].

## 7. Antibiotic Therapy

A definite overview of randomized controlled trials on antibiotic administration in non-pregnant females with lower UTI [40] demonstrated the therapeutic benefits of immediate antibiotic administration. Antibiotic use resulted in complete symptom resolution in 61.8% of patients compared to 25.7% of non-users. The authors concluded that the evidence available from studies was of high quality according to the GRADE (Grade of Recommendations, Assessment, Development, and Evaluations) system of evidence [41]. The use of antibiotics further led to urine culture sterilization in 90% of cases, compared to 33.3% in non-users; the relevant evidence was considered of moderate quality. Moreover, antibiotic use led to significantly fewer relapses/recurrences (15.8% versus 41.6% in non-users; evidence of moderate quality). The authors could not evaluate the effect of antibiotic use on the risk of pyelonephritis development, since the pyelonephritis episodes recorded in the included studies were too rare. As expected, the use of antibiotics resulted in an increased incidence of side effects, with 19.2% of antibiotic users versus 12.9% of non-users reporting side effects (evidence of moderate quality). Diarrhea was the commonest side effect recorded.

When choosing a proper antibiotic for uncomplicated lower UTIs—what is more, when choosing it empirically—one should take into account the collateral damage induced in community antimicrobial resistance trends as well as the already existing community antimicrobial resistance. Since this empirical administration is supported by available evidence, as already outlined in Section 5, certain principles should be set in order to maximize the results and minimize undesirable individual and community effect. Two of the universally suggested first-line antibiotics, fosfomycin and nitrofurantoin, exert minimal resistance pressure on the intestinal flora, and thus minimal ecological resistance pressure on the community: a fact reflected in the minimal development of resistance against these two agents [42]. Another important parameter in the empirical choice of antibiotic is the current *E. coli* community resistance pattern, with a rule (based on expert consensus reproduced in the relevant Infectious Diseases Society of America (IDSA) and European Society for Microbiology and Infectious Diseases 2011 guidelines [19]) of excluding any antibiotic with a reported resistance exceeding 20% in the specific country/region of origin/residence.

Having a common therapeutic framework adjusted to these pre-requisites would be extremely helpful for all frontline health practitioners to serve as navigators (assessing who is in need of treatment), therapists (suggesting the proper antibiotic), and gatekeepers (ensuring that the proper antibiotic is administered, in the proper dose, for the proper period). This is not always the case, though, Greece being a prime example: there are three different guidelines for the treatment of UTIs, originating from three different official state or scientific organizations. The guidelines of the Ministry of Health [43], which are incorporated in official prescription protocols too (and can thus be considered "binding" for those prescribing), include as first-line options a single 3 gr dose of fosfomycin, or a 100 mg t.i.d. dose of nitrofurantoin for 5–7 days, or a dose of 400 mg b.i.d. or 200 mg t.i.d. of pivmecillinam for 5 days. Second-line choices include fluoroquinolones, the combination of amoxicillin and clavulanate, and cephalosporins.

The official guidelines of the National Organization for Medicines/EOF [44] are rather similar, proposing the use of first-line nitrofurantoin or fosfomycin in the same regimens as the Ministry of Health, as well as the use of pivmecillinam in the 400 mg b.i.d. regimen. These guidelines, though, further include the use of trimethorpim/sulfamethoxazole in a dose of 160/800 mg b.i.d. One should note here that *E.coli*'s resistance against trimethoprim/sulfamethoxazole remains systematically above 20% in Greece, according to WHONET data, with a more recent official result for the second semester of 2021 [45]; thus, its use in empirically treating an uncomplicated UTI in Greece should be discouraged. In this report, based on isolates from urine cultures from outpatients, trimethoprim/sulfamethoxazole resistance reached 23.4%. Resistance to ciprofloxacin,

ampicillin/sulbactam, and nalidixic acid also exceeded the threshold of 20%, and a percentage exceeding 20% was also reported for resistant and intermediate-sensitivity isolates against amoxicillin–clavulanate and cephalothin.

Finally, the guidelines of the Hellenic Society for Infectious Diseases [11] suggest a drastically different approach, with first-line regimens including 500/125 mg of amoxicillin/clavulanate t.i.d. for 5–7 days, 3-day fluoroquinolone regimens, 50–100 mg of nitrofurantoin q.i.d. for 5–7 days, and 160/800 mg of trimethoprim/sulfamethoxazole b.i.d. for 3 days. These guidelines are in stark contrast not only with the other state-originating Greek guidelines, but also with most international scientific and regulatory guidelines, which coincide in proposing nitrofurantoin, fosfomycin, pivmecillinam, and trimethoprim/sulfamethoxazole (the latter where local resistance patterns allow) as first-line therapy.

A discrepancy observed when comparing various international organization guidelines regards treatment duration: according to the British NICE guidelines, administering a 3-day regimen achieves a favorable efficacy/side effect balance; on the other hand, it has been persuasively demonstrated that 3-day regimens, when compared to 5-day ones, are inferior in terms of incidence of relapse in the following 4–10 weeks [11,46]. Clinical studies are warranted and could focus on whether a subset of patients would safely benefit from a 3-day regimen without an increased relapse rate, possibly taking into account age and past history of UTIs.

The use of fluoroquinolones is generally discouraged for two reasons: one is the ecological effect of their use on antimicrobial resistance development trends. But their side effect/benefit balance is also an important parameter that is often neglected. Regarding resistance, one should note that already in Greece, according to the most recent report from the European Centre of Disease Control with data from 2021, more than 30% of nosocomial *E. coli* strains were resistant to quinolones [47]. The aforementioned WHONET data for the second semester of 2021 further exhibited a disappointing 20.3% of resistant community-sampled *E. coli* strains [44]. Despite regulatory body advice on limiting the use of fluoroquinolones [48], their use has actually been on the rise in Greece as well as in certain Eastern European countries, including Bulgaria, Poland, Hungary, and Slovakia [49].

In recent years, though, a more important reason for minimizing fluoroquinolone use has been stressed by regulatory bodies such as the EMA: its significant, and potentially non-reversible, side effects, especially an increased risk of aortic aneurysm rupture and dissection [50]. Furthermore, one should not ignore the lowering of the threshold of developing convulsions (an effect that may be more dramatic if concurrently used with NSAIDs that may act in a similar fashion), the potential impact on cardiac rhythm indices, which may be complex in patients with relevant pre-existing pathology or on relevant medication [51], and, finally, the well-known induction of tendinitis (particularly when co-administered with corticosteroids) [52]. Fluoroquinolones are contra-indicated in glucose-6-phosphate dehydrogenase (G6PD) deficiency.

When choosing the first-line therapeutic regimen, other factors that weigh in include drug–drug interactions and side effects. The antimicrobials that are considered first-line options have a generally similar, benign side effect profile [53]. Both trimethoprim/sulfamethoxazole and nitrofurantoin are contra-indicated in cases of G6PD deficiency. Trimethoprim/sulfamethoxazole is also contra-indicated in porphyria patients, as well as in cases of folate deficiency. The use of trimethoprim/sulfamethoxazole should preferably be avoided in individuals that concurrently use medications causing hyperkalemia [54] and hyponatremia (potentially caused by the antibiotic also). Similarly, nitrofurantoin should not be the first choice in individuals with underlying conditions predisposing to peripheral neuropathy [55] (as in patients with long-standing diabetes mellitus, patients with vitamin B complex deficiency, and patients with electrolyte disorders).

Technical issues regarding the optimal antibiotic administration should also be considered: for example, nitrofurantoin's activity in narrow urine pH levels (with toxicity due to accumulation in acidic environment and inactivation in alkaline environment) [56] and the need for dose adjustment or discontinuation when the glomerular filtration rate is impaired.

A similar important issue regards fosfomycin use, a single dose of which should be administered on an empty stomach and ideally when the patient's bladder is empty (and preferably before going to bed at night) [57].

How feasible is it to skip antibiotics and go into a 48 h "wait and observe" mode? The British guidelines include this option (in contrast to all Greek and most other international guidelines), considering it feasible for females 18–44 years old without a history of past UTIs. Further studies are needed though on the subject.

Attempting a synopsis, when one synthesizes the guidelines from the Greek Ministry of Health, the National Organization for Medicines, the European Urological Association, and the British NICE guidelines, taking into account the local Greek community resistance patterns, an optimal therapeutic approach, as the one summarized in Table 1, can be reached: nitrofurantoin, fosfomycin, and pivmecillinam can be considered suitable first-line choices (with individual patient parameters allowing for further selection between the three). According to the IDSA grading system [58], all three choices achieve an excellent grade ranging from A1 to A2, indicating a satisfying level of evidence for their use, originating from either more than one randomized trial of superior quality or more than one well-designed non-randomized clinical trial, cohort study, case–control study originating in more than one study centers, or multiple time period patient series. According to the GRADE rating system [41], these treatment options are strongly recommended due to the presence of high-quality evidence, belonging to the first grade (which equals desirable results far outperforming side effects and evidence of efficacy arising from properly performed randomized controlled trials or exquisite results arising from unbiased observational trials). Such a recommendation is not expected to be overturned by future studies.

**Table 1.** Suggested therapeutic first-line regimens for empirical treatment of uncomplicated UTI based on a synthesis of national and international guidelines on par with data from country (Greece) resistance patterns.

| Suggested Antibiotic | Dose |
| --- | --- |
| Nitrofurantoin | 100 mg t.i.d. for 5 days |
| Fosfomycin | 3 gr, single dose |
| Pivmecillinam | 400 mg b.i.d. for 5 days |

Regarding therapeutic follow-up, one should re-evaluate symptoms and diagnosis and refer to an expert if there is no response after 48 h of treatment. Relapse may warrant a different therapeutic regimen, while chronic recurrent UTI necessitates the performance of urine cultures. If urine culture has already been performed in the first place and the result is available after the initiation of empirical antimicrobial therapy, the prescribed antibiotic may continue to be administered if there is an adequate clinical response, irrespective of antibiogram results, since there are often differences between in vitro evidence and in vivo efficacy.

Asymptomatic bacteriuria does not warrant treatment, apart from pregnancy (where a 10–20% possibility of evolution to ascending infection and pyelonephritis has been traditionally reported) and prior to localized surgical interventions. The relevant EAU guidelines [7], as well as the IDSA ones [59], note that the available evidence on treatment of asymptomatic bacteriuria in pregnancy is based on older studies with numerous methodological drawbacks; both guidelines state that treatment of asymptomatic bacteriuria in pregnancy warrants further studies due to the results of a recent large, well-designed Dutch study that has raised the question of whether screening for asymptomatic bacteriuria (and subsequent treatment) is actually needed in all pregnancies [60]: the study showed that pregnant females with asymptomatic bacteriuria that were not treated with nitrofurantoin were more prone to developing pyelonephritis compared to pregnant women that received the antibiotic, but the overall prevalence of pyelonephritis was extremely low (2.4%). Furthermore, the presence of untreated bacteriuria did not exert an increased

likelihood of preterm birth compared to pregnant women without bacteriuria. On the other hand, evidence for the treatment of asymptomatic bacteriuria prior to invasive urologic interventions is more definite, with an EAU meta-analysis of available trials underlining a significant benefit despite considering these trials of poor quality [61].

There is limited evidence at present (which should be remedied) regarding urinary tract infections in transgender individuals.

Similarly, there is limited evidence about the utility of self-diagnostic tests, where available, both in terms of sensitivity and improvement of early diagnosis and in terms of specificity and risk of over-diagnosing [62].

## 8. Recurrent UTIs

### 8.1. UTI Relapse

UTI relapse can be due to improper choice of antibiotic treatment or inadequate duration of treatment. It has also been associated with frequent sexual contact [8].

### 8.2. Chronically Recurrent UTIs

Chronically recurrent UTIs with a short time period between episodes may be etiologically linked with sexual contact, frequent changes in sexual partners, use of contraceptive and spermicidal devices, genetic predisposition, urinary incontinence or residual post-void urine, diabetes mellitus, immunocompromising conditions, and anatomical or functional urinary tract disorders (that may have run unnoticed until then) [63,64].

In dealing with a chronically recurrent UTI, certain everyday practices may augment, such as the need for perineal hygiene achieved by wiping front to back, repeat urination in order to ensure that there is no residual urine volume (particularly after sexual contact), use of well-fitting underwear, and proper hydration. Regarding the latter, one study has demonstrated that by increasing daily water intake to 1.5 lt, one can lower the incidence of recurrences in the following year [65].

Recurrences can be prevented, or become rarer, with prophylactic administration of antibiotics or alternative approaches: prophylactic antibiotics can be administered daily or post-coitally, with no efficacy differences between these two approaches. Chemoprophylaxis may use either fosfomycin, in a single dose repeated every 10 days for a total of 6 months, or nitrofurantoin, in a 50–100 mg single daily dose, or trimethoprim/sulfamethoxazole in a 40/200 mg single daily dose (for the latter, provided that a pre-requisite urine culture has isolated a pathogen susceptible to trimethoprim/sulfamethoxazole). The Hellenic Society for Infectious Diseases suggests a daily therapeutic administration for 6–12 months or a post-coital administration of either nitrofurantoin (50 mg for post-coital dose) or trimethoprim/sulfamethoxazole in the aforementioned dosing regimens or the use of cephalosporins (250 mg of cefaclor or 125 mg of cephalexin in single daily doses), or even the use of fluoroquinolones (which should of course be discouraged for the reasons analyzed in Section 7). TEAU though, limits its suggestion of cephalosporin use only to pregnancy and proposes the daily use of 50–100 mg of nitrofurantoin or the use of 3 g of fosfomycin every 10 days or the use of 100 mg of trimethoprim daily [7].

A number of non-antibiotic approaches have been evaluated in clinical practice. Intravaginal estrogens have been shown to be less effective but with a superior side effect profile compared to chemoprophylaxis. Immunotherapy with oral regimens, including OM89 and AV140, has limited support from the available literature [66]. Probiotics have demonstrated significant efficacy in recurrence prevention, at least for certain types of lactobacilli [67]; a recent study evaluated the use of vaginal probiotics based on lactobacilli, either in isolated use or in parallel with oral probiotics based on lactic acid bacteria and bifidobacteria, and resulted in a strong prophylactic outcome [68]. On the other hand, the use of D-mannose (which acts by interfering with the binding of *E. coli* to bladder epithelial cells) was considered of inconclusive efficacy in a Cochrane database review of its use [69]. Cranberry extracts have long been used in recurrence prophylaxis, similarly with inconclusive evidence [70]. Regarding the use of methenamine, which in the acidic urine

environment hydrolyzes to formaldehyde (which is the active substance), was considered to have weak and statistically insignificant efficacy in a 2021 meta-analysis [71], but a subsequent randomized control trial showed non-inferiority compared to chemoprophylaxis [72]. Methenamine is not extensively available (not in Greece for example).

## 9. Conclusions

Uncomplicated lower UTIs will continue to be a prevalent community infectious disease that can be readily, safely, and effectively diagnosed and treated by frontline healthcare professionals without the need for a sophisticated diagnostic workup. The diagnostic, differential diagnostic, and therapeutic approach should be common for all health professionals, and should be based on a favorable benefit/side effect balance for the patient, on individual patient characteristics, and local microorganism resistance patterns, as well as on the overall benefit/pressure on resistance evolution balance. The authors aimed to develop such a framework (Figure 1), common for all frontline healthcare workers (general practitioners, clinical microbiologists, pharmacists) who may be the first to encounter a patient with symptoms compatible with UTI, based on national and international guidelines and taking into account Greek bacterial resistance trends. There is an abundance of adjuvant, non-antibiotic approaches suggested in the literature or even considered theoretically effective by certain international regulatory bodies, yet there is limited quality evidence in the relevant literature regarding such interventions, underlining the need for further well-designed evaluation of such approaches.

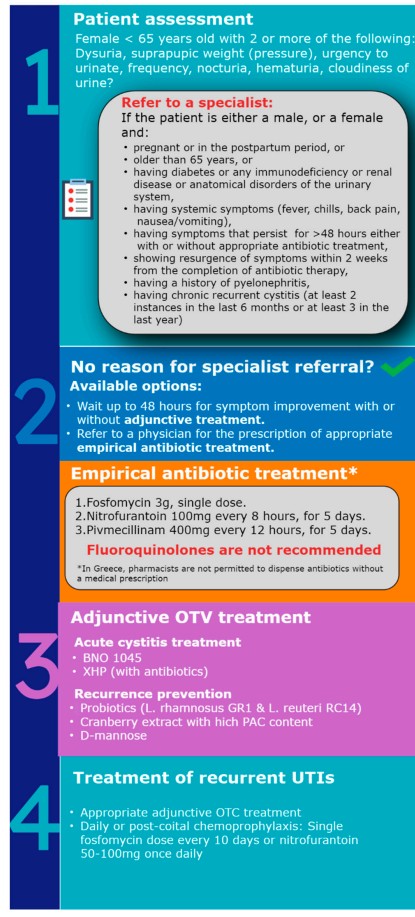

**Figure 1.** An algorithm of first-line health professionals' suggested approach to a patient with symptoms compatible with urinary tract infection.

**Author Contributions:** Conceptualization, methodology, software, validation, formal analysis, investigation, resources, data curation, writing—original draft preparation, writing—review and editing, visualization, supervision, project administration, D.K. and G.P. All authors have read and agreed to the published version of the manuscript.

**Funding:** This research received no external funding.

**Conflicts of Interest:** The authors declare no conflict of interest.

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
