# Peer review of "Lower Urinary Tract Infections: An Approach for Greek Community Health Practitioners"

_2813-9054, doi:10.3390/amh69020007_

Round 1

Reviewer 1 Report

Comments and Suggestions for Authors

I have read with interest the manuscript submitted by the authors since UTI represent an important concern worldwide. 

- once an abbreviation is defined (such as UTI), the authors should use it furthermore in text

- row 52 - "antibiotics with significant resistance patterns in the specific community" - an antibiotic does not have a resistance pattern, only the microorganism.

- rows 124-125: "As already mentioned though, the term “absence of any symptom” can be debatable in elderly individuals." - already mentioned where?

- a great amount of information provided is not supported by a reference (for example rows 103-135, 214-250, 371-435 do not have a single reference).

- row 309 - Chapter 4??

The manuscript is not well structured and seems to be just a compilation of other articles, without proper editing and even mentioning the sources. It contains multiple English errors and is hard to understand.

Comments on the Quality of English Language

extensive editing is needed.

Author Response

  1. abbreviations have been appropriately used in the text, as suggested.
  2. row52, "microorganisms" used instead of antibiotics, as correctly suggested. 
  3. row 124-125: was mentioned in rows 109-110 in the original manuscript (now, in the non-tracked revised manuscript, it's 140-141 and 123-124 respectively). 
  4.  30 additional references have been added, focusing in particular in the suggested paragraphs, we apologize for our initial inadequacy
  5. row309, "chapter" corrected to "section"
  6. extensive language revision has been made throughout the manuscript.
  7. Regarding the structure of the manuscript: we aimed at synthesizing available guidelines and recommendations, in a manner that may be accessible to non-specialists practitioners. We follow a typical structure of introduction-basic knowledge- definitions- diagnosis- treatment approach- outcomes, but we never aimed at developing new guidelines (it would be extremely arrogant to aim at something like this, anyway). Of course we have to base on certain official recommendations, but we have not focused strictly on one (and have not copy-pasted anything!), instead aiming at addressing the common ground in the most important of them (EAU and NICE and IDSA are obvious candidates), and the most topical of them. We apologize for omitted references and have extensively corrected this issue. 

Reviewer 2 Report

Comments and Suggestions for Authors

Dear Authors,

The manuscript is interesting, especially for health practitioners dealing with UTIs in Greece. Greece occupies the first place in Europe concerning antibiotic consumption.

The manuscript is well-written and organized. In my opinion, the authors should try to be more concise. There is a lot of information in the article without any references. Also, the authors should include recent information about the AMR of uropathogens in Greece or at least in the Balkans. 

l. 8 Uncomplicated UTIs can be safely diagnosed by history and clinical examination

l. 46-55 is hard to understand; please reformulate

page 2, which contains only one reference

l. 83-92 needs some references

l. 103-135 needs some references

l. 137-142 needs some references

7. Recurrent UTIs - please include recommendations from EAU guidelines

Please include conclusions

Comments on the Quality of English Language

The English is fine.

Author Response

  1. we have added 30 more references, going from 41 to 71, particularly in paragraphs that were unreferenced- we apologize for our initial error. Information on European resistance data have been limitedly added, since most ECDC reports focus on hospital samples and not community ones. We have added additional information about the Greek resistance profile though. The AAC Stapleton article from 2020 that aimed at reporting the worldwide E. coli resistance patterns shows that most studies are from more than 15 years ago, thus not representative of the current situation.
  2. row8: changed in the revised version to "diagnosed through history"
  3. 46-55 paragraph has been extensively rewritten (is now rows 59-70 in the non tracked revised manuscript)
  4. references have been added both for page 2, as well as the suggested subsequent paragraphs
  5. the EAU guidelines for recurrent UTIs have been added as suggested. 
  6. a final Conclusions paragraph has been added as suggested.

Reviewer 3 Report

Comments and Suggestions for Authors

UTI may seem a simple and tedious subject, but all the clinicians know that in fact is a complex and problematic issue. The authors choose a difficult task, to synthetize the recommendations of three different Guidelines form the same country and they do it in an organised manner.

There are some issues that require attention, but overall the article has a lot of merits. 

1. Line 5-6: "Uncomplicated UTIs refer to non-recurrent infections"

Actually uncomplicated infections may be also recurrent. The misleading statement is present only in the abstract and not in the body of the article. Please review and correct.

2. Although this article is a review, some paragraphs (including some that refer to studies published in the literature) lack references. Such paragraphs are lines 180-183, lines 208-211, lines 216-222, lines 223-225, lines 397-407, lines 442-455

3. Lines 344-345 "This consensus further aims at salvaging quinolones, by not using them when available alternative, equally or superiorly efficient, treatment options exist."

One of the most important recommendation against using quinolones when other options exist is based on their potential adverse effects. In line 355 authors state "Their side effects/ benefit balance has also been a subject of debate." but I think this is somehow an understatement. I think the issue of quinolones side effects deserve a comment in the article.

4. Lines 419-420 "Asymptomatic bacteriuria does not warrant treatment, apart from pregnancy .., and prior to localized surgical interventions in children"

According to most of the guidelines antibiotic treatment of asymptomatic bacteriuria is warranted in patients scheduled for surgical interventions on the urinary tract (and this recommendation is stronger than in pregnant women, where the importance of the treatment should be weighted by the urologist and the obstetritician). Please comment.

Comments on the Quality of English Language

Only minor corrections should be made.

In line 51, I do not understand what "over-jealous antibiotic choices mean". Is it maybe over-zealous? please review and correct if necessary.

Line 454: maybe the word "chapter" should be replaced by the word "section"?

Author Response

  1. line 5-6: we have erased the "non-recurrent" part, as correctly suggested.
  2. we have added 30 more references, going from 41 to 71, focusing in particular in the suggested paragraphs from the present and other reviewers. We apologize for our initial error. 
  3.  We have redesigned the discussion on fluoroquinolones, focusing particularly in side-effects, with relevant references added, and with strengthening their importance, as mentioned, according to reviewer suggestion. 
  4.  we have added a small paragraph discussing how the quality of available evidence on treating asymptomatic bacteriuria in pregnancy and pre-urological surgery differs, and how this is viewed from both EAU and IDSA. We also added a brief mention of the Dutch study from 2015 raising the issue of whether all ASBs in pregnancy need treatment. 
  5. line 51, corrected to over-zealous, as suggested, apologies for the initial error
  6. line 454, "section" has been used instead of "chapter" as correctly suggested. 

Round 2

Reviewer 1 Report

Comments and Suggestions for Authors

I appreciate the author's efforts in addressing my comments. The quality of the manuscript has improved.

- there is still a great amount of information provided, which is not supported by a reference.

- I never suggested that the authors should have provided new guidelines for UTI treatment, I just had comments regarding the structure of the manuscript.

- rows 142-147 - please provide the reference and present the differences between reinfection/recurrence.

According to the EAU guideline, rUTIs are defined as "Recurrences of uncomplicated and/or complicated UTIs, with a frequency of at least three UTIs/year or two UTIs in the last six months"

- rows 457-459 - and what was the answer to that question?

rows 460-462 - references to support the statement.

Best regards

Comments on the Quality of English Language

- there are still multiple edits required (both for English language and punctuation)

Author Response

  • there is still a great amount of information provided, which is not supported by a reference.
    We have further added a reference (as specified in a subsequent comment) and have further linked some statements of the manuscript to already existing references.
  • I never suggested that the authors should have provided new guidelines for UTI treatment, I just had comments regarding the structure of the manuscript.
    We have already responded about the structure of the manuscript- we consider our approach a logical one, following typical steps of medical literature. We have already stated the reason the manuscript was created. We understand that Reviewer 1 considers such an approach a useless one, given existing literature on the subject, and we respect their view. Yet, in a country with three different official guidelines on UTI, that furthermore is a EU champion in community antimicrobial resistance, we still consider our approach a useful step in enhancing awareness on the proper UTI approach for all frontline health practitioners.
  • rows 142-147 - please provide the reference and present the differences between reinfection/recurrence.
    We have changed the term "recurrence" to "relapse", in order to avoid overlap with chronic recurrent UTIs. We have added the relevant literature for the definition of relapse. We have added a short clarification on what is considered as relapse and what as reinfection, according to the relevant literature.
  • According to the EAU guideline, rUTIs are defined as "Recurrences of uncomplicated and/or complicated UTIs, with a frequency of at least three UTIs/year or two UTIs in the last six months"
    We have rewritten the definition of chronic recurrent UTI according to EAU description.
  • rows 457-459 - and what was the answer to that question?
    We have added an analytical presentation of the findings of this study and have rewritten the paragarph in order to underline its importance for both IDSA and EAU guidelines.
  • rows 460-462 - references to support the statement.
    We have added the relevant reference of the meta-analysis. Apologies for omitting it in the first place.
  • Comments on the Quality of English Language
    - there are still multiple edits required (both for English language and punctuation)
    The article has been extensively rewritten in the first revision. Further improvements, less extensive, have been made. Errors in punctuation are unfortunate and we apologize for them. 

Reviewer 2 Report

Comments and Suggestions for Authors

Dear Authors,

Congratulations on your work; the manuscript has improved.

Comments on the Quality of English Language

The English is fine

Author Response

thank you for the comments, no further changes were indicated

Reviewer 3 Report

Comments and Suggestions for Authors

All the requested modifications were made. The article can be published in this form.

Author Response

thank you for the comments, no further changes were requested

Round 3

Reviewer 1 Report

Comments and Suggestions for Authors

In my opinion, the manuscript could be published after some English and punctuation editing,

I congratulate the authors for their efforts.

Best wishes.

Comments on the Quality of English Language

minor

Author Response

Thank you for the comments.

Further language and grammar corrections were made, and all punctuation errors were corrected